# CD70-Targeted Micelles Enhance HIF2α siRNA Delivery and Inhibit Oncogenic Functions in Patient-Derived Clear Cell Renal Carcinoma Cells

**DOI:** 10.3390/molecules27238457

**Published:** 2022-12-02

**Authors:** Noah Trac, Hyun Seok Oh, Leila Izzy Jones, Randy Caliliw, Shinji Ohtake, Brian Shuch, Eun Ji Chung

**Affiliations:** 1Department of Biomedical Engineering, University of Southern California, Los Angeles, CA 90089, USA; 2Institute of Urologic Oncology, University of California, Los Angeles, CA 90095, USA; 3Division of Nephrology and Hypertension, Department of Medicine, Keck School of Medicine, University of Southern California, Los Angeles, CA 90089, USA; 4Department of Medicine, Norris Comprehensive Cancer Center, Keck School of Medicine, University of Southern California, Los Angeles, CA 90089, USA; 5Department of Medicine, Eli and Edythe Broad Center for Regenerative Medicine and Stem Cell Research, Keck School of Medicine, University of Southern California, Los Angeles, CA 90089, USA; 6Division of Vascular Surgery and Endovascular Therapy, Department of Surgery, Keck School of Medicine, University of Southern California, Los Angeles, CA 90089, USA; 7Mork Family Department of Chemical Engineering and Materials Science, University of Southern California, Los Angeles, CA 90089, USA

**Keywords:** clear cell renal cell carcinoma, micelle, HIF2α, siRNA, CD70, nanoparticle

## Abstract

The majority of clear cell renal cell carcinomas (ccRCCs) are characterized by mutations in the Von Hippel–Lindau (*VHL*) tumor suppressor gene, which leads to the stabilization and accumulation of the HIF2α transcription factor that upregulates key oncogenic pathways that promote glucose metabolism, cell cycle progression, angiogenesis, and cell migration. Although FDA-approved HIF2α inhibitors for treating VHL disease-related ccRCC are available, these therapies are associated with significant toxicities such as anemia and hypoxia. To improve ccRCC-specific drug delivery, peptide amphiphile micelles (PAMs) were synthesized incorporating peptides targeted to the CD70 marker expressed by ccRCs and anti-HIF2α siRNA, and the ability of HIF2α-CD27 PAMs to modulate HIF2α and its downstream targets was evaluated in human ccRCC patient-derived cells. Cell cultures were derived from eight human ccRCC tumors and the baseline mRNA expression of *HIF2A* and *CD70*, as well as the HIF2α target genes *SLC2A1*, *CCND1*, *VEGFA*, *CXCR4*, and *CXCL12* were first determined. As expected, each gene was overexpressed by at least 63% of all samples compared to normal kidney proximal tubule cells. Upon incubation with HIF2α-CD27 PAMs, a 50% increase in ccRCC-binding was observed upon incorporation of a CD70-targeting peptide into the PAMs, and gel shift assays demonstrated the rapid release of siRNA (>80% in 1 h) under intracellular glutathione concentrations, which contributed to ~70% gene knockdown of HIF2α and its downstream genes. Further studies demonstrated that knockdown of the HIF2α target genes *SLC2A1*, *CCND1*, *VEGFA*, *CXCR4*, and *CXCL12* led to inhibition of their oncogenic functions of glucose transport, cell proliferation, angiogenic factor release, and cell migration by 50–80%. Herein, the development of a nanotherapeutic strategy for ccRCC-specific siRNA delivery and its potential to interfere with key oncogenic pathways is presented.

## 1. Introduction

Clear cell renal cell carcinoma (ccRCC) comprises 75–80% of all kidney cancers, with 50% of high-risk patients relapsing and a five-year survival rate of less than 20% for those with metastatic disease [1,2,3]. The current clinical standard for the systemic treatment of ccRCC consists of combination therapy using an immune checkpoint inhibitor (ICI), such as an anti-PD1 antibody with either a CTLA-4 antibody or a VEGFR tyrosine kinase inhibitor (TKI). However, adverse events (AEs) related to treatment toxicity remain common [4]. For example, in the CLEAR phase III clinical trial in which 352 patients were treated with a combination of the PD1 antibody pembrolizumab and the TKI lenvatinib, grade 3–4 AEs were observed in 82% of patients, resulting in 68% of patients receiving dose reductions and another 13% discontinuing treatment [5]. Thus, the development of efficacious alternative therapies that minimize patient AEs remains a major need [6,7].

To boost treatment efficacy and minimize toxicity, therapies targeted to specific molecular targets of ccRCC have been recently explored. For example, 70–90% of ccRCCs are characterized by loss of function of the Von Hippel–Lindau (*VHL)* tumor suppressor gene, which suppresses the activity of and destabilizes hypoxia-induced factors (HIFs) [8,9,10,11,12]. This has generated interest in the development of therapies targeting HIFs, especially HIF2α, which when overexpressed upregulates oncogenes including *SLC2A1*, *CCND1*, *VEGFA*, *CXCR4*, and *CXCL12* that support cancer progression through increased glucose transport and glycolysis, cell cycle progression, angiogenesis, and cell migration [13,14,15,16]. To that end, in 2021, a selective inhibitor of HIF2α (belzutifan) was FDA approved for VHL disease and is in further clinical development for metastatic ccRCC. Despite efficacy, belzutifan also suppresses HIF2α/erythropoetin in hepatocytes leading to anemia in 90% of treated patients [17,18,19,20].

To limit off-target toxicity, nanocarriers with tunable size, charge, and surface properties can be utilized to increase the specificity of drug delivery to target tissues [21,22,23,24]. For example, studies by our and other groups have shown that nanoparticles less than 100 nm in size achieve the greatest kidney accumulation [25,26,27,28]. In this proof-of-concept study, we developed peptide amphiphile micelles (PAMs) that are 8–20 nm in size and incorporate siRNA targeted to HIF2α. In order to increase the specificity of HIF2α siRNA delivery to ccRCC cells, PAMs were functionalized with a targeting ligand capable of binding to CD70, a transmembrane protein reported to be highly expressed in ccRCCs [29,30]. The chosen targeting ligand is a 13-mer peptide derived from CD27, the binding partner of CD70 [31]. After successfully developing HIF2α-CD27 PAMs, we evaluated their siRNA release profile under intracellular glutathione concentrations. Then, to assess the efficacy of HIF2α-CD27 PAMs, we tested micelle binding to human patient tissue-derived ccRCC cells in vitro and evaluated their ability to inhibit cancer cell glucose transport, proliferation, release of angiogenic factors, and migration. Overall, the in vitro efficacy of HIF2α-CD27 PAMs in downregulating multiple oncogenic mechanisms and their clinical potential for ccRCC gene therapy are provided.

## 2. Results and Discussion

### 2.1. Synthesis of HIF2α-CD27 PAMs and Characterization of siRNA Loading and Release

Based on our earlier studies developing nanoparticles that accumulate in the kidneys in vivo, peptide amphiphile micelles (PAMs) that are typically less than 20 nm in diameter and are able to penetrate the glomerular filtration barrier were developed for ccRCC [32]. To enhance ccRCC specificity, we incorporated a peptide targeted to the CD70 transmembrane protein expressed by ccRCCs (CD27 peptide). CD27 peptides or thiolated HIF2α siRNAs were conjugated to DSPE-PEG2000-maleimide and self-assembled into HIF2α-CD27 PAMs under aqueous conditions at a 99:1 peptide:siRNA ratio (Figure 1A and Appendix A) [33]. Non-targeting PAMs were synthesized using scrambled CD27 (scrCD27) peptides (Appendix A). Transmission electron microscopy (TEM) images confirmed micelles are uniform and spherical in morphology (Figure 1B), and dynamic light scattering (DLS) measurements showed the HIF2α-CD27 PAMs to be 13.8 ± 0.4 nm in diameter (Appendix A). Using zeta potential measurements, the surfaces of the CD27 PAMs became more negatively charged upon incorporation of the anionic siRNA as expected, from −28.8 ± 6.0 to −40.5 ± 13.6 (Figure 1C) [34,35].

To evaluate HIF2α siRNA incorporation into PAMs, a gel shift assay was performed to compare HIF2α-CD27 PAMs (500 ng siRNA, 13.3 μg PAM) and free HIF2α siRNA (500 ng siRNA) migration [36]. As shown in Figure 1D, following 90 min of electrophoresis at 50 V, >95% of HIF2α-CD27 PAMs (lane i) remained in the well, demonstrating that the HIF2α siRNA was successfully incorporated into the micelle. In contrast, the free HIF2α siRNA (lane ii) readily migrated down the gel. In addition, incorporation of the siRNA into micelles was found to protect the nucleic acid cargo from degradation, as approximately 90% of HIF2α-CD27 PAMs incubated with RNase-treated FBS for 1 h were retained in the well (lane iii), while unconjugated siRNA incubated with RNase-treated FBS for 1h showed no signal in the gel (lane iv), indicating that the siRNA had been degraded into smaller oligonucleotides and migrated off the gel [37,38].

Upon internalization into the cell, HIF2α-CD27 PAMs experience intracellular levels of glutathione (GSH) up to 10 mM, which can be several orders of magnitude higher than extracellular GSH [39,40,41,42,43]. To characterize the release of therapeutic siRNA from the PAMs under intracellular conditions, additional gel electrophoresis assays were performed following PAM incubation with 10 mM GSH for up to 24 h [44]. siRNA release was calculated by comparing the band intensity in the well (non-released siRNA) to the band intensity further down the gel (released siRNA) (Appendix A). As found in Figure 1E, >70% of the siRNA was released after 1 h incubation with intracellular levels of GSH, indicating that HIF2α-CD27 PAMs are capable of rapidly releasing their payload after internalization into target cells. In contrast, HIF2α-CD27 PAMs incubated at extracellular concentrations of GSH (1 μM) released <25% of siRNA cargo after 24 h.

### 2.2. HIF2α and Downstream Genes Are Upregulated in Patient-Derived ccRCC Cells

To determine if HIF2α siRNA delivery represents a viable therapeutic strategy for ccRCC patients, baseline mRNA expression of *HIF2A*, *CD70*, and cancer-supporting downstream targets regulated by HIF2α such as *SLC2A1*, *CCND1*, *VEGFA*, *CXCR4*, and *CXCL12* were evaluated in immortalized HK-2 renal proximal tubule cells and eight patient-derived primary ccRCC cell cultures (Figure 2A–G) [45,46,47,48,49,50]. Patient characteristics for each of these cultures, including tumor grade, size, and stage are listed in Table 1 [51,52,53].

As found in Figure 2, in general, the ccRCC samples had significantly higher baseline mRNA expression of *HIF2A*, *SLC2A1*, *CCND1*, *VEGFA*, and *CXCR4* compared to the HK- 2 kidney epithelial cell line (*p* < 0.05). Although overall mRNA expression of *CD70* and *CXCL12* in the ccRCC samples was greater than that of the control HK-2 cells, the differences were not statistically significant which is attributed to the high variation in gene expression across the ccRCC samples. Analyzing the individual samples, 5 of the 8 (63%) patient-derived samples had significantly higher *CD70* mRNA expression compared to the HK-2 control cells. Additionally, 7 of the 8 (88%) samples had increased *CXCL12* mRNA expression (Table 2). Out of all samples, Sample 5 which was derived from a patient with metastatic ccRCC, had significantly higher mRNA expression for all the target genes compared to the HK-2 control cells (Table 3). Because of this, we hypothesized that Sample 5 would be the most susceptible to CD70-targeted HIF2α siRNA therapy and continued with this sample for subsequent in vitro studies.

### 2.3. CD70-Targeting Micelles Bind to Patient-Derived ccRCC Cells In Vitro

To evaluate the specificity of HIF2α-CD27 PAMs for binding to CD70^+^ ccRCC cells, we incubated FITC-labelled CD27 PAMs with patient-derived ccRCC cells in vitro for 30 min (Figure 3A). Immunofluorescent antibody staining confirmed CD70 expression and that CD27 PAMs colocalized with CD70 (Pearson’s colocalization coefficient *R* = 0.54). In contrast, minimal binding was observed in cells treated with scrCD27 PAMs (Figure 3B). To further confirm the specificity of HIF2α-CD27 PAMs for binding to CD70^+^ ccRCC cells, the binding of CD27 PAMs was also assessed following a 30 min incubation of the ccRCC cells with anti-CD70 polyclonal antibodies (Figure 3C). As expected, blocking CD70 prior to PAM treatment significantly decreased PAM binding to levels comparable to the scrCD70 control. To quantify these results, we repeated these experiments with cells grown in microtiter plates and measured the fluorescence signal and found that, in agreement with the qualitative microscopy images, CD27 PAMs had significantly more binding than scrCD27 PAMs, and that this increased binding was not observed if the cells were pre-treated with anti-CD70 (Figure 3D). The specificity of CD27 PAMs for CD70 is also apparent through our binding studies with the HK-2 cell line, which has been reported to have minimal CD70 expression [54]. When incubated with HK-2 cells for 30 min, no differences in micelle binding were observed between CD27 PAMs, scrCD27 PAMs, or CD27 PAMs after anti-CD70 incubation (Appendix A).

In addition to patient-derived cells, binding studies were also performed ex vivo using formalin-fixed, paraffin-embedded (FFPE) ccRCC tumor tissue sections by treating sections with fluorescently labeled 10 μM PAMs for 1 h (Figure 3E,F). Similar to Figure 3A–D, we confirmed that CD70 was expressed by IHC, and that CD27 PAMs bound to ccRCC tissue sections with greater specificity compared to scrCD27 PAMs.

### 2.4. HIF2α-CD27 PAM Treatment Reduces HIF2α mRNA Expression In Vitro

To evaluate the efficacy of siRNA delivery with CD70-targeted PAMs, patient-derived ccRCC cells were treated with HIF2α-CD27 PAMs, HIF2α-scrCD27 PAMs, scrHIF2α-CD27 PAMs, free HIF2α siRNA, or PBS for 48 h at siRNA concentrations of 500 nM. The MTS proliferation assay was used to confirm that the transfection protocol was non-toxic to the cells (>90% viability, Appendix A). Then, *HIF2A* and *CD70* mRNA expression were assayed through qRT-PCR (Figure 4). Cells treated with HIF2α-CD27 PAMs were observed to have reduced *HIF2A* expression (31.2 ± 5.6%) relative to the PBS control (*p* < 0.05). Additionally, the *HIF2A* knockdown mediated by HIF2α-CD27 PAMs was larger than that of non-targeting HIF2α-scrCD27 PAMs (54.8 ± 9.6%, *p* < 0.05) and free HIF2α siRNA (65.9 ± 19.6%, *p* < 0.05), suggesting that incorporation of the HIF2α siRNA into a CD70-targeted micelle increased its uptake into the ccRCC cells and thus enhanced gene silencing, which corroborates the results observed in the binding studies in Figure 3. Similarly, HIF2α-CD27 PAM treatment also reduced *CD70* mRNA expression compared to the PBS control (24.1 ± 12.0), as well as the other treatment groups (*p* < 0.05), as also reported by other groups that found *CD70* expression to be correlated to HIF2α expression [54,55].

### 2.5. HIF2α-CD27 PAMs Inhibit In Vitro Glucose Transport and Proliferation by Reducing SLC2A1 and CCND1 Expression

To evaluate if the gene knockdown observed in HIF2A extended to its downstream targets as well, we conducted qRT-PCR on several downstream oncogenes, starting with *SLC2A1*, which controls glucose uptake [56]. *SLC2A1* modulates GLUT1, the transmembrane glucose transporter strongly expressed in ccRCC that contributes to increased glucose metabolism, ATP generation, and cell growth [57,58]. To evaluate the effect of HIF2α-CD27 PAM treatment on GLUT1-mediated glucose transport, patient-derived ccRCC cells were treated with HIF2α-CD27 PAMs, HIF2α-scrCD27 PAMs, scrHIF2α-CD27 PAMs, free HIF2α siRNA, or PBS for 48 h (500 nm siRNA), and then *SLC2A1* mRNA expression was assayed through qRT-PCR (Figure 5A). HIF2α-CD27 PAMs significantly reduced SLC2A1 mRNA expression compared to HIF2α-scrCD27 PAMs, HIF2α siRNA, PBS-treated cells (*p* < 0.05), demonstrating the potential of CD70-targeted micelles for siRNA delivery for ccRCC. To test if the significant knockdown at the mRNA level impacted cell phenotype, a glucose uptake assay was performed 24 h following siRNA treatment (Figure 5B). HIF2α-CD27 PAM treatment reduced glucose uptake by 47.8%, 32.8%, and 36.4% relative to PBS, HIF2α-scrCD27 PAMs, and free HIF2α siRNA, respectively (*p* < 0.05).

Given the ability of HIF2α-CD27 PAMs to restrict glucose metabolism in ccRCC cells, the effect of HIF2α-CD27 PAM treatment directly on ccRCC cell proliferation was examined. mRNA expression of *CCND1*, a downstream target of HIF2α that regulates cell cycle progression, was evaluated 48 h following HIF2α-CD27 PAM treatment (500 nM siRNA) [59,60]. As shown in Figure 5C, HIF2α-CD27 PAM treatment reduced *CCND1* expression by 76.8% (*p* < 0.05) and was observed to be more efficacious in knocking down *CCND1* than all other groups (*p* < 0.05). Next, the effect of HIF2α-CD27 PAM treatment on ccRCC cell proliferation was evaluated 3 d and 5 d after treatment using the MTS proliferation assay. While there was only a 15% reduction in cell proliferation in the HIF2α-CD27 PAM-treated cells after 3 d (Appendix A), the anti-proliferative effect of micelle treatment was more pronounced 5 d after treatment, with a 54.4% reduction in cell proliferation relative to the PBS-treated control (Figure 5D, *p* < 0.05) and was more potent than HIF2α-scrCD27 PAMs and free HIF2α siRNA (*p* < 0.05). These functional studies demonstrate the potential of HIF2α-CD27 PAMs to directly modulate metabolic pathways and their therapeutic ability to inhibit ccRCC cell growth and proliferation.

### 2.6. Culture Medium Collected from HIF2α-CD27 PAM-Treated ccRCC Cells Shows Anti-Angiogenic Properties

As mentioned, anti-angiogenic TKIs are often used as the clinical standard for the ccRCC treatment. Thus, to understand how HIF2α-CD27 PAMs alter the induction of angiogenesis in ccRCC tumors, we first evaluated the mRNA expression of the angiogenic *VEGFA* gene following PAM treatment and found that HIF2α-CD27 PAMs reduced *VEGFA* expression by approximately 75% compared to the PBS group (*p* < 0.05 vs. PBS, HIF2α-scrCD27 PAMs, and free HIF2α siRNA, Figure 6A). Then, we collected and incubated conditioned culture medium from treated patient-derived ccRCC cells with human umbilical vein endothelial cells (HUVECs) for 72 h. As shown in Figure 6B, HUVECs cultured with conditioned medium from HIF2α-CD27 PAM-treated cells grew 40% slower than cells cultured with conditioned medium from PBS-treated cells, likely due to the reduced production and release of angiogenic growth factors in cells treated with HIF2α-CD27 PAMs (Figure 6B) [61,62]. These assays demonstrate the multiple, therapeutic benefits of HIF2α-CD27 PAMs that may aid to slow vascularization and growth of tumors in future in vivo studies.

### 2.7. HIF2α-CD27 PAM Treatment Reduces Patient-Derived ccRCC Cell Migration and Wound Closure

Finally, to further verify the therapeutic effects of HIF2α-CD27 PAM treatment on ccRCC cell migration and mobility, testing was performed through wound healing assays on patient-derived ccRCC cells (Figure 7A). Cells were treated with HIF2α-CD27 PAMs for 48 h before an artificial wound was introduced to the monolayer and cell migration was imaged up for to 24 h. As shown in Appendix A and Figure 7A, HIF2α-CD27 PAM treatment slowed wound closure by approximately 80% compared to the PBS-treated control after 24 h (*p* < 0.05). HIF2α-scrCD27 PAM treatment slowed wound closure by approximately 50% (*p* < 0.05), while free HIF2α siRNA treatment did not have any significant effect on wound closure. These results are consistent with the qRT-PCR data, which show ~65% knockdown of chemotactic markers *CXCR4* and *CXCL12* following 48 h of HIF2α-CD27 PAM treatment, and lesser knockdown from the HIF2α-scrCD27 PAM and free HIF2α treatments (Figure 7B). As such, our studies collectively demonstrate that HIF2α-CD27 PAMs are capable of binding specifically to ccRCC cells and exert a therapeutic effect through the modulation of HIF2α-related oncogenic genes and function.

## 3. Conclusions

In summary, we report the successful synthesis and characterization of peptide amphiphile micelles incorporating siRNA targeted to HIF2α and CD70-targeting peptides and their ability to exert multiple anti-tumor effects in patient-derived ccRCC cells and tumor tissues. We confirmed that incorporating CD70-targeting peptides to PAMs increased micelle binding to both patient-derived ccRCC cells in vitro and on ex vivo tissue sections. We report that HIF2α-CD27 PAMs were consistently more efficacious than both free siRNA and non-targeted PAMs in silencing gene expression and inhibiting their oncogenic functions in glucose metabolism, cell cycle progression, angiogenesis, and cell migration, highlighting the importance of both siRNA encapsulation and molecular targeting, and demonstrating the potential of HIF2α-CD27 PAMs for ccRCC-specific drug delivery. As such, future studies will evaluate the in vivo efficacy of HIF2α-CD27 PAMs using a ccRCC mouse model towards evaluating its clinical potential.

## 4. Materials and Methods

### 4.1. Materials and Cells

Amino acids were purchased from Gyros Protein Technologies (Uppsala, Sweden) and Sigma Aldrich (St. Louis, MO, USA). PEGylated lipids were purchased from Avanti Lipids (Alabaster, AL, USA). Cy7 mono-*N*-hydroxysuccinimide (NHS) ester was purchased from Lumiprobe (Hunt Valley, MD, USA). DSPE-PEG2000-FITC was purchased from Creative PEGWorks (Durham, NC, USA). HIF2α and control RNA duplexes were purchased from IDT Technologies (Coralville, IA, USA). Antibodies were purchased from Thermo Fisher (Waltham, MA, USA). Cell lines were purchased from the American Type Culture Collection (ATCC, Manassas, VA, USA). Cell culture reagents were purchased from Gibco (Waltham, MA, USA) and Sigma Aldrich. De-identified, human clear cell renal cell carcinoma (ccRCC) tissue samples (1 tumor per patient) were collected on an IRB-approved protocol, measured, evaluated for grade and stage, and then generously donated from UCLA Urology/PI-Dr. Brian Shuch.

### 4.2. Synthesis of HIF2α-CD27 Peptide Amphiphile Micelles

CD27 (CRKAAQCDPCIPG) and scrambled CD27 (CQGPRACKADPIC) peptides were synthesized on a Rink amide resin using Fmoc-mediated solid phase peptide synthesis with an automated peptide synthesizer (PS3, Gyros Protein Technologies) and N-capped with an acetyl group. Peptide deprotection and cleavage from the resin was performed with two 2 h incubations with a 94:2.5:2.5:1 vol% mixture of trifluoroacetic acid (TFA):water:ethanedithiol:tri-isopropylsilane (TIS) [63]. To control conjugation to DSPE-PEG2000-maleimide, peptides were synthesized with TFA-stable acetamidomethyl (Acm) groups protecting the side chains of the non-terminal cysteines, which were deprotected later with 10 eq. mercury acetate [64]. Peptides were purified using reverse-phase high performance liquid chromatography (RP-HPLC, Prominence, Shimadzu, Columbia, MD, USA) on a Luna C18 column (Phenomenex, Torrance, CA, USA) at 55 °C using HPLC-grade water and acetonitrile (Fisher Scientific, Hampton, NH, USA) supplemented with 0.1% formic acid. Peptide purity was characterized using matrix-assisted laser desorption/ionization time-of-flight mass spectrometry (MALDI-TOF-MS, Bruker, MA, USA). The expected *m*/*z* is 1403 (Appendix A).

Pure peptides were then mixed with a 10% molar excess of DSPE-PEG2000-maleimide in water before adjustment of the pH to 7.2 [65]. The reaction was nitrogen purged and allowed to stir for 3 d before purification through RP-HPLC using a Luna C4 column (Phenomenex). Acm groups were removed using 10 eq. of mercury acetate, and the pure peptide amphiphiles were desalted through RP-HPLC and characterized through MALDI-TOF-MS. The expected *m*/*z* is 4343 *m*/*z* (Appendix A). Fluorescent DSPE-PEG2000-cy7 amphiphiles were synthesized by dissolving DSPE-PEG2000-amine in a 0.1 M sodium bicarbonate buffer, and then mixing it with 3-fold molar excess of cy7 NHS ester solubilized in dimethyl formamide (DMF, 10% of reaction volume). The reaction was allowed to shake overnight before purification using a Luna C4 column and characterization via MALDI-TOF-MS. HIF2α siRNA (sense: 5′-CUUGCAGUUUUACUAAAACACUGAA-3′, antisense: 5′-UUCAGUGUUUUAGUAAAACUGCAAGGG-3′) thiolated on the 5′ end of the sense strand was conjugated to DSPE-PEG2000-maleimide through a thioether bond by mixing the siRNA and lipid in nuclease-free water adjusted to pH 7.2 overnight [66]. Conjugation of DSPE-PEG2000-HIF2α siRNA amphiphiles was confirmed through the gel electrophoresis assays described in 4.5.

Micelles were self-assembled from peptide and siRNA amphiphiles through thin-film hydration. Peptide amphiphiles were dissolved in methanol, and a nitrogen stream was used to evaporate the methanol, leaving a thin lipid film that was then hydrated with nuclease-free water or PBS containing the siRNA amphiphiles, gently sonicated and vortexed, and heated to 40 °C for 30 min. Unless otherwise stated, siRNA and fluorescent amphiphiles were incorporated into micelles at 1 mol% and 10 mol%, respectively.

### 4.3. Dynamic Light Scattering (DLS) and Zeta Potential

Micelles were suspended at 50 μM in 1 mM NaCl in a folded capillary zeta cell for size, polydispersity, and zeta potential measurements using a Zetasizer Ultra instrument (Malvern Panalytical, Malvern, UK) at room temperature [67]. N = 3.

### 4.4. Transmission Electron Microscopy (TEM)

Micelles were imaged using a FEI Talos F200C microscope (Thermo Fisher). An amount of 5 μL of a 25 μM micelle solution in water was placed directly onto a 400 mesh Carbon Type-B TEM grid (Ted Pella, Redding, CA, USA), washed with water, and then stained with 2% uranyl acetate before a final wash [68]. Grids were left overnight in the dark prior to imaging.

### 4.5. Gel Electrophoresis Assay

HIF2α-CD27 (500 ng siRNA, 13.3 μg CD27) PAMs or free HIF2α siRNA (500 ng siRNA) was loaded into the wells of a 2% (w/v) agarose gel containing 0.5 μg/mL ethidium bromide and RNA migration was observed using a ChemiDoc XRS+ imaging system (Bio-Rad, Hercules, CA, USA) after the application of 50 V for 90 min [69]. For RNase treatment, samples were incubated for 1 h with 50 μg/mL RNase A (Thermo Fisher) in 5% fetal bovine serum (FBS) prior to gel electrophoresis.

### 4.6. Characterization of siRNA Release

An amount of 300 μM HIF2α-CD27 PAMs was incubated with 10 mM or 1 μM glutathione (GSH) for up to 24 h in nuclease-free water and then diluted 5-fold in nuclease-free water immediately before gel electrophoresis was performed under the conditions detailed in Section 4.5. Relative band intensity was quantified using ImageJ and used to calculate % siRNA release. N = 3.

### 4.7. Isolation and Culture of Patient-Derived ccRCC Cells

Freshly resected tumor samples procured at time of partial or radical nephrectomy were transported on ice before dicing finely using a pair of scalpels. Afterwards, samples were digested in low-serum culture media containing 2 mg/mL Type-II collagenase and 2% penicillin-streptomycin overnight at 37 °C, passed sequentially through 70 μm and 40 μm filters, and then washed twice in fresh RPMI before seeding onto collagen-coated tissue culture plates or flasks [70]. Cells were cultured in RPMI supplemented with 10–20% FBS and 1% penicillin–streptomycin.

### 4.8. In Vitro Micelle Binding to Patient-Derived ccRCC Cells

Approximately 200,000 patient-derived ccRCC cells or HK-2 cells were seeded onto glass coverslips placed in the wells of a 6-well plate and allowed to adhere overnight. Then, 50 μM FITC-labeled CD27 or scrambled CD27 micelles were incubated with the cells for 30 min at 37 °C and fixed with fresh 4% paraformaldehyde (PFA) and blocked with 0.3 M glycine and 5% normal goat serum in PBS-Tween-20 (PBST). After blocking, cells were incubated with anti-CD70 primary antibodies (1:200) overnight at 4 °C and then AF594 secondary antibodies (1:500) for 1 h at room temperature. Cells were then counterstained with 2 μg/mL DAPI and mounted to Superfrost Plus microscope slides (Fisher) using Vectamount aqueous mounting media (Vector Laboratories, Burlingame, CA, USA) and sealing with clear nail polish. Microscope slides were allowed to set overnight in the dark at room temperature before imaging on an LSM 880 confocal microscope. In addition to microscopy, 10,000 cells were also seeded into black 96-well plates, incubated with 50 μM FITC-labelled micelles for 30 min, washed, and then fluorescence was measured using a Synergy H4 Hybrid microplate reader (Agilent, Santa Clara, CA, USA, n = 6). To block cell surface CD70, cells were incubated with anti-CD70 (1:20) for 30 min at 37 °C prior to micelle incubation.

### 4.9. Immunohistochemical (IHC) Staining and PAM Binding to ccRCC Patient Tissue Sections

ccRCC patient tumor samples were formalin-fixed and paraffin-embedded (FFPE) and sectioned at 10 μm. Sections were then stained with hematoxylin and eosin or prepared for immunofluorescent (IF) antibody staining. Briefly, for IF staining, tissue sections were heated to 100 °C in pH 6 citrate buffer for 15 min, blocked with 5% normal goat serum in TBS-Tween-20 (TBST) for 30 min, incubated with anti-CD70 (1:200) primary antibodies at 4 °C overnight, washed with TBST and then incubated for 1 h with Alexa Fluor 594-conjugated secondary antibodies (1:500) at room temperature. Following antibody staining, tissue sections were washed and then incubated with 10 μM CD27 micelles for 1 h before washing and counterstaining with 2 μg/mL DAPI. Coverslips were mounted using Prolong Gold mounting medium (Thermo Fisher) and allowed to cure overnight before imaging using a Zeiss LSM 880 confocal microscope (Zeiss, Oberkochen, DE).

### 4.10. In Vitro Transfection and mRNA Expression of Patient-Derived ccRCC Cells

An amount of 100 μL micelles or free siRNA was added to the wells of a 12-well plate at a concentration of 5 μM RNA, and then 900 μL of a solution containing approximately 100,000 cells in 5% FBS/RPMI was added (n = 5). Cells were incubated for 48 h at 37 °C, after which the cells were lysed, and RNA was isolated using the mRNeasy Kit from Qiagen (Hilden, DE). cDNA was synthesized using the RT^2^ First Strand Kit according to manufacturer instructions (Qiagen). Expression of *HIF2A*, *CD70*, *SLC2A1*, *CCND1*, *VEGFA*, *CXCR4*, *CXCL12*, and *GAPDH* was evaluated through real-time PCR with primer assays and the RT^2^ SYBR Green qPCR Mastermix (Qiagen) using a CFX384 Touch Real-Time PCR Detection System (Bio-Rad Laboratories), according to the manufacturer’s instructions. Fold-change in mRNA expression was calculated using the delta-delta Ct method.

### 4.11. MTS Assay

In vitro micelle biocompatibility was evaluated by incubating micelles or siRNA with patient-derived RCC or HK-2 cells for 48 h in 5% FBS/RPMI before adding MTS reagent (10% *v*/*v*, Abcam, Cambridge, UK) and incubating for 1 h. Afterwards, absorbance at 490 nm was quantified using a microtiter plate reader. Cell % viability was determined following blank subtraction by normalizing the treatment values to PBS-treated control wells. The effect of siRNA transfection on cell proliferation was determined by transfecting cells according to Section 4.10, replacing the transfection solution with fresh complete culture media, and then incubating for an additional 48–120 h before adding the MTS reagent for subsequent absorbance measurements (n = 5).

### 4.12. Glucose Uptake Assay

Cells (5000 in number) were plated in the wells of a 96-well plate (n = 4) and treated with HIF2α-CD27 PAMs, HIF2α-scrCD27 PAMs, scrHIF2α-CD27 PAMs, HIF2α siRNA, or PBS at an siRNA concentration of 500 nM for 48 h at 37 °C. Afterwards, the cell culture media was refreshed, and the cells were allowed to grow unperturbed for 24 h before evaluation of glucose uptake using the Glucose Uptake-Glo assay (Promega, Madison, WI, USA) according to the manufacturer’s instructions. Glucose uptake was evaluated via luminescence measurements from a Synergy H4 Hybrid Microplate Reader.

### 4.13. Collection of ccRCC-Conditioned Cell Culture Medium and In Vitro Culture and Growth of Endothelial Cells

ccRCC cells were transfected as detailed in Section 4.10, then cell culture medium was refreshed. After 24 h, conditioned culture medium was collected and stored at 4 °C [71] and 5000 HUVECs were seeded into the wells of a 96-well plate and allowed to adhere and grow over 48 h. Then, conditioned culture medium was diluted 1:1 in a 4:1 mixture of serum-free RPMI and endothelial cell growth medium. This mixture was added to the HUVECs, and the cells were allowed to grow for 72 h before evaluation of cell proliferation via MTS assay (n = 5) [72].

### 4.14. Wound Healing Assay

Cells (50,000 in number) were seeded into the wells of a 48-well plate (n = 4) and cultured for 5 d to attain a confluent monolayer. Then, cells were treated with HIF2α-CD27 PAMs, HIF2α-scrCD27 PAMs, scrHIF2α-CD27 PAMs, HIF2α siRNA, or PBS according to Section 4.10. After treatment, the media in the wells was replaced with fresh culture media and a 200 uL pipette tip was used to create a single scratch on the surface of the wells, and brightfield images of the cells were taken with a Leica DMi8 microscope (Leica, Wetzlar, DE) 0, 3, 6, 9, 12, and 24 h after scratching [73]. A scalpel blade was used to mark the wells to ensure microscope images were taken in the same field-of-view for each timepoint. Wound closure was measured as the area of the scratch using ImageJ.

### 4.15. Statistics

Data are expressed as mean ± SD. Statistical analysis between two groups was performed using a Student’s *t*-test. Comparisons between three or more groups were performed through analysis of variance (ANOVA) followed by post hoc Dunnett’s test for multiple comparisons. A *p*-value ≤ 0.05 was considered to be statistically significant.

## Figures and Tables

**Figure 1 molecules-27-08457-f001:**
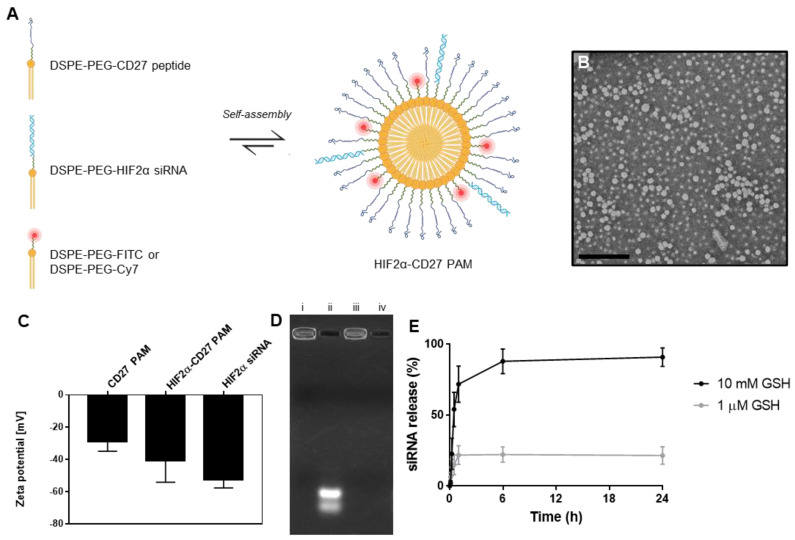
Physicochemical characterization of HIF2α-CD27 PAMs. (**A**) Schematic of HIF2α-CD27 PAM self-assembly. (**B**) Transmission electron micrograph of HIF2α-CD27 PAMs. Scale bar = 200 nm. (**C**) Zeta potential of CD27 PAMs before and after HIF2α siRNA incorporation. N = 3. (**D**) Gel electrophoresis assay demonstrating incorporation of siRNA into PAMs at 1 mol% and protection of siRNA from RNAse-mediated degradation after 1 h. Lanes: (i) HIF2α-CD27 PAMs, (ii) HIF2α siRNA, (iii) HIF2α-CD27 PAMs incubated with RNAse-treated FBS for 1 h, (iv) HIF2α siRNA incubated with RNAse-treated FBS for 1 h. (**E**) Gel electrophoresis assay with PAMs up to 24 h after GSH treatment demonstrate rapid siRNA release following exposure to intracellular GSH levels (1 mM) and minimal siRNA release (<25%) after extracellular GSH exposure (1 μM). N = 3.

**Figure 2 molecules-27-08457-f002:**
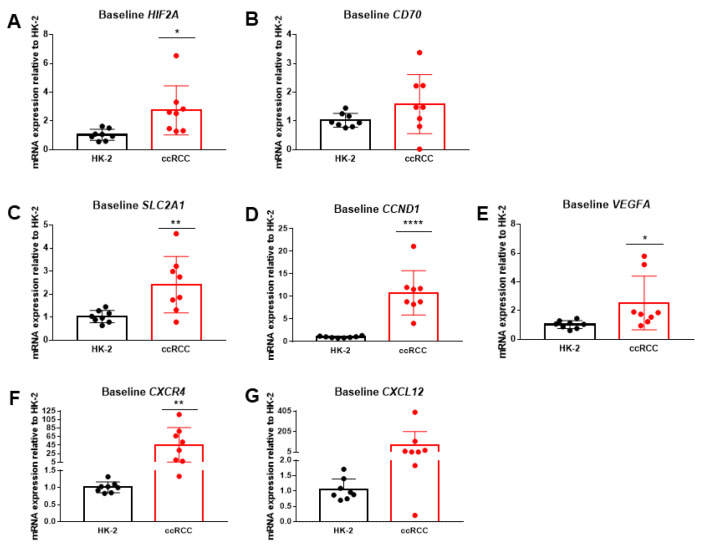
Baseline mRNA expression in patient-derived ccRCC cell cultures. Patient-derived ccRCC cells have increased (**A**) *HIF2A*, (**B**) *CD70*, (**C**) *SLC2A1*, (**D**) *CCND1*, (**E**) *VEGFA*, (**F**) *CXCR4*, (**G**) *CXCL12* mRNA expression compared to HK-2 kidney epithelial cells. N = 8. * *p* < 0.05, ** *p* < 0.001, **** *p* < 0.0001.

**Figure 3 molecules-27-08457-f003:**
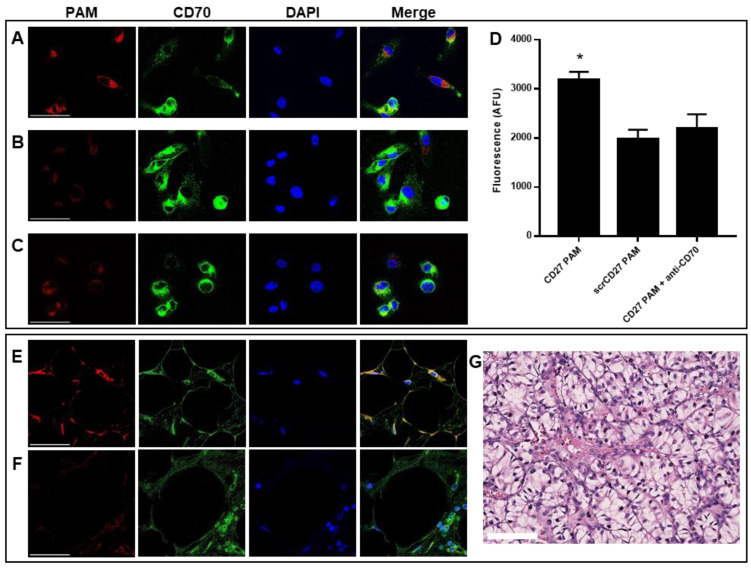
In vitro binding to patient-derived ccRCC cells and to tumor tissue sections. Confocal microscopy images of patient-derived ccRCC cells that were immunostained for CD70 (green), counterstained with DAPI (blue), and incubated for 30 min with 50 μM FITC-labeled micelles (red). (**A**) CD27 PAMs have increased binding to ccRCC cells in vitro compared to (**B**) scrCD27 PAMs. (**C**) CD27 PAM binding is reduced after pre-incubation with anti-CD70 antibodies, confirming specificity of CD27 PAMs for CD70. (**D**) CD27 PAMs have significantly increased binding compared to scrCD27 PAMs, but not if target cells are pre-treated with anti-CD70. N = 6. * *p* < 0.05. (**E**) ccRCC tumor tissue sections incubated for 1 h with 10 μM CD27 micelles have increased nanoparticle signal compared to (**F**) scrCD27 micelles. Scale bar = 50 μm. (**G**) Representative H&E-stained ccRCC tumor tissue section. Scale bar = 100 μm.

**Figure 4 molecules-27-08457-f004:**
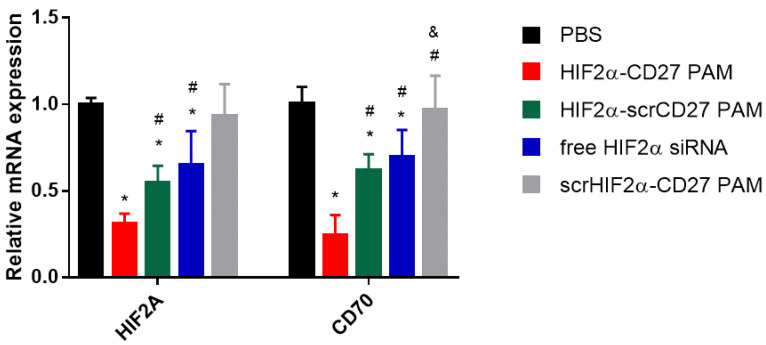
mRNA expression of HIF2A (left) and CD70 (right) following 48 h treatment with HIF2α-CD27 PAMs. The 48 h HIF2α-CD27 PAM treatment (500 nM siRNA) significantly reduced HIF2A and CD70 mRNA expression compared to non-targeted PAM and free siRNA controls in patient-derived ccRCC cells. N = 5. * *p* < 0.05 relative to PBS. ^#^
*p* < 0.05 relative to HIF2α-CD27 PAMs. ^&^
*p* < 0.05 relative to HIF2α-scrCD27 PAMs.

**Figure 5 molecules-27-08457-f005:**
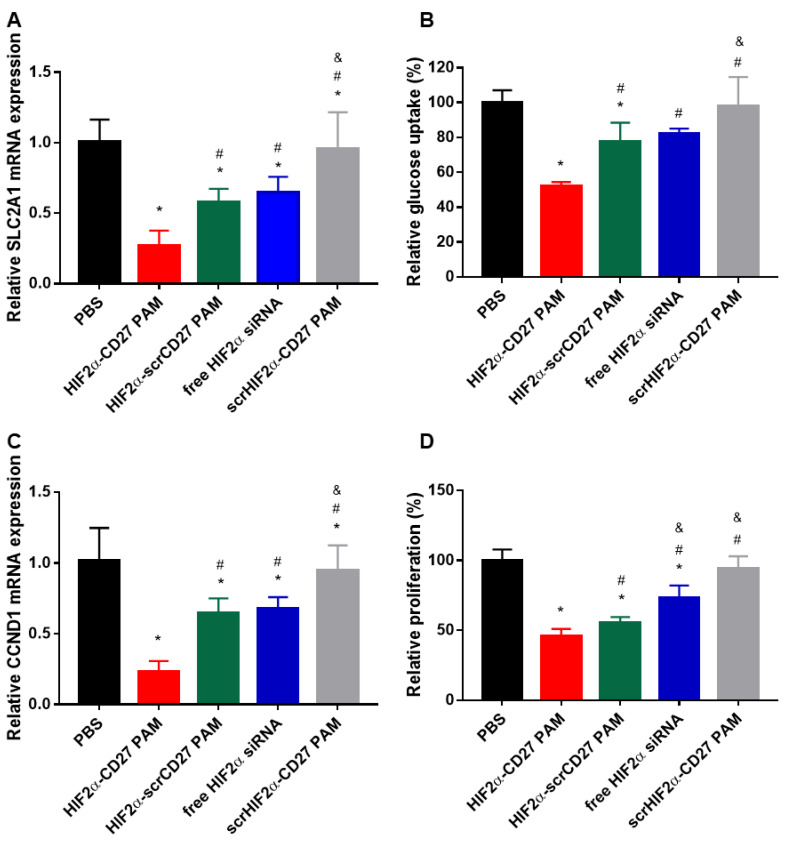
Gene knockdown and functional effects of HIF2α-CD27 PAM treatment. 48 h HIF2α-CD27 PAM treatment (500 nM siRNA) reduces (**A**) SLC2A1 mRNA expression and (**B**) glucose transport relative to non-targeted PAMs, free siRNA, and PBS in patient-derived ccRCC cells. HIF2α-CD27 PAMs also reduce (**C**) CCND1 mRNA expression and (**D**) ccRCC proliferation 5 d following treatment. N = 4–6. * *p* < 0.05 relative to PBS. ^#^
*p* < 0.05 relative to HIF2α-CD27 PAMs. ^&^
*p* < 0.05 relative to HIF2α-scrCD27 PAMs.

**Figure 6 molecules-27-08457-f006:**
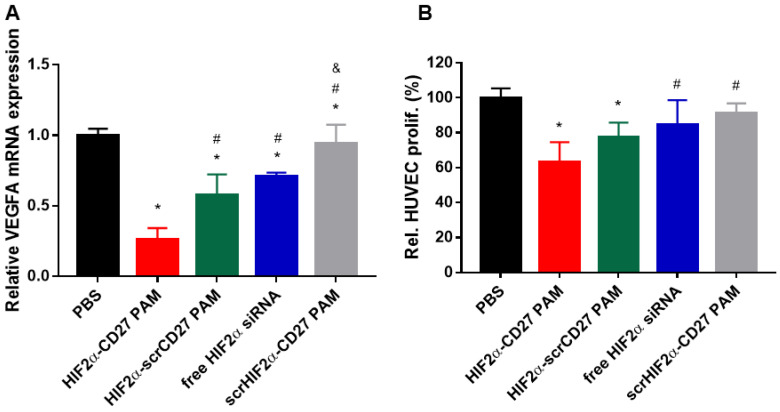
48 h HIF2α-CD27 PAM treatment (500 nM siRNA) reduces (**A**) VEGFA mRNA expression in patient-derived ccRCC. (**B**) HUVECs cultured with HIF2α-CD27 PAM-conditioned culture medium have reduced proliferation relative to PBS-conditioned culture medium. N = 5. * *p* < 0.05 relative to PBS. ^#^
*p* < 0.05 relative to HIF2α-CD27 PAMs. ^&^
*p* < 0.05 relative to HIF2α-scrCD27 PAMs.

**Figure 7 molecules-27-08457-f007:**
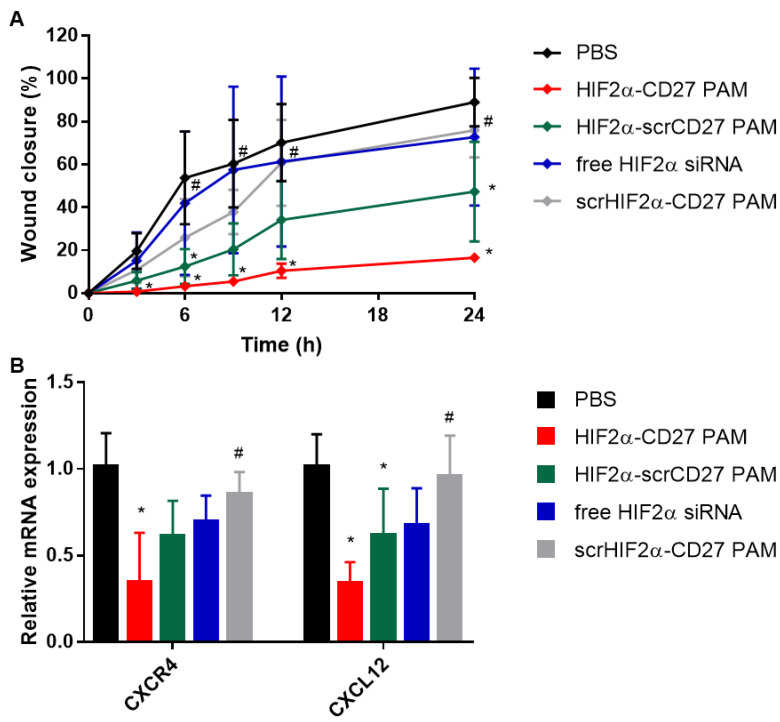
Effect of HIF2α-CD27 PAMs on ccRCC migration. (**A**) The 48 h HIF2α-CD27 PAM treatment (500 nM siRNA) slows wound closure of ccRCC cells over 24 h compared to PBS and free siRNA treatment. (**B**) The 48 h HIF2α-CD27 PAM treatment (500 nM) reduces mRNA expression of CXCR4 (left) and CXCL12 (right). N = 4 or 5. * *p* < 0.05 relative to PBS. ^#^
*p* < 0.05 relative to HIF2α-CD27 PAMs.

**Table 1 molecules-27-08457-t001:** ccRCC patient characteristics.

Sample	Grade	Tumor Size (cm)	Stage
1	3	2.5	I
2	2	6.3	III
3	3	4.7	III
4	3	11	III
5	2	3.6	IV *
6	2	4.1	I
7	3	15.6	III
8	4	13.5	III

* Metastatic ccRCC.

**Table 2 molecules-27-08457-t002:** Patient-derived ccRCC cells with higher HIF2α-related gene expression compared to HK- 2 cells.

Gene	N (%)
*HIF2A*	5/8 (63%)
*CD70*	5/8 (63%)
*SLC2A1*	5/8 (63%)
*CCND1*	8/8 (100%)
*VEGFA*	7/8 (88%)
*CXCR4*	7/8 (88%)
*CXCL12*	7/8 (88%)

*p* < 0.05.

**Table 3 molecules-27-08457-t003:** mRNA expression of individual ccRCC samples, normalized to HK-2 cells.

	Sample							
Gene	1	2	3	4	5	6	7	8
*HIF2A*	6.54 * ± 1.99	3.31 * ± 0.39	1.28 ± 0.39	1.47 ± 0.18	2.84 * ± 0.72	1.32 ± 0.19	2.61 * ± 0.50	2.52 * ± 0.67
*CD70*	0.81 ± 0.25	1.08 ± 0.14	2.23 * ± 0.36	2.22 * ± 0.45	3.38 * ± 0.24	1.48 * ± 0.22	0.02 * ± 0.00	1.48 * ± 0.18
*SLC2A1*	1.32 ± 0.08	4.63 * ± 1.10	1.86 * ± 0.77	1.75 * ± 0.17	3.22 * ± 0.68	2.75 * ± 0.47	0.79 ± 0.11	2.99 * ± 0.02
*CCND1*	8.83 * ± 1.23	8.82 * ± 1.02	11.75 * ± 1.55	11.58 * ± 0.28	8.27 * ± 1.22	4.04 * ± 0.80	21.09 * ± 3.35	12.00 * ± 2.54
*VEGFA*	1.91 * ± 0.18	5.20 * ± 0.95	0.95 ± 0.19	1.75 * ± 0.14	5.79 * ± 1.40	1.86 * ± 0.21	1.23 * ± 0.16	1.55 * ± 0.14
*CXCR4*	116.90 * ± 75.61	66.95 * ± 25.37	33.25 * ± 8.98	7.40 * ± 3.89	10.45 * ± 4.19	52.47 * ± 18.75	1.33 ± 0.55	77.82 * ± 14.67
*CXCL12*	393.77 * ± 219.76	0.21 * ± 0.13	1.83 * ± 0.44	11.59 * ± 3.00	4.06 * ± 1.85	20.73 * ± 7.77	109.59 * ± 18.41	4.38 * ± 2.76

N = 3. * *p* < 0.05 vs. HK-2 cells.

## Data Availability

The data presented in this study are available upon request to the corresponding author.

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
