# Peer review of "CD70-Targeted Micelles Enhance HIF2α siRNA Delivery and Inhibit Oncogenic Functions in Patient-Derived Clear Cell Renal Carcinoma Cells"

_molecules, 2022, doi:10.3390/molecules27238457_

Round 1
Reviewer 1 Report
1. CD70 and CD27 are rarely described in the introduction,which I suggest to describe in more detail.
2. The reduction of toxicity is mentioned many times in the introduction, but it is not reflected in the experiment and conclusion. I suggest the authors add toxicity-related experiments in this manuscript.
3. The scale of B in Figure 1 should be presented in the figure.
4. For the study of the release of siRNA, n=1 is not strictly prohibited, I suggest n=3.
5. Table 3 should add SD values.
Reviewer 2 Report
In the authors list, please include the number of the author adscription like Noah Trac1 instead of Noah Trac1
In abstract and in Materials and Methods sections: Please the genes names such as SLC2A1, CCND1, VEGFA, CXCR4, and CXCL12 in italics.
Line 157: genes name in italics.
Line 22: The Von Hippel Lindau (VHL) tumor suppressor gene, please VHL in italics.
Line 140: Is there a statistical difference between tumor stage or grade and the targets genes levels? I suggest to authors performed an analysis to determine if the tumor grade or stage affected the mRNA expression levels and how the initial level of CD70 has an effect over the micelle’s treatment. Is there an SNP of CD70 that could have an effect over the micelles binding capacity?
Line 189: In vitro (italics)
Line 169: please define scrCD27 as scrambled CD27
Reviewer 3 Report
Noah Trac and co-workers reported the peptide amphiphile micelles (PAMs) incorporating siRNA targeted to HIF2α. Further functionalized with a targeting ligand derived from CD27. Characterized the micelles and evaluated their clinical potential. Overall, the study is well conducted and well reported. This paper deserved to be published in Molecules after the following changes.
1. Report the DLS size distribution data for the micelles. Report span values.
2. Report the number of observations for each panel and comparisons must be clearly defined.
3. Release study methodology is not sufficient. Elaborate. Did you replenish the medium?
4. Table tumor sizes don’t have SD values why is that? Make this clear.
5. Almost no references in methodology. Are these methods your own? Please cite the relevant references.
